# Identification of Seven Additional Genome Segments of Grapevine-Associated Jivivirus 1

**DOI:** 10.3390/v15010039

**Published:** 2022-12-22

**Authors:** Thierry Candresse, Laurence Svanella-Dumas, Armelle Marais, Flora Depasse, Chantal Faure, Marie Lefebvre

**Affiliations:** INRAE, UMR BFP, University of Bordeaux, CS20032, CEDEX, 33882 Villenave d’Ornon, France

**Keywords:** jivivirus, divided genome, non-coding regions, high-throughput sequencing, genomic RNA, supplementary RNA

## Abstract

Jiviruses are a group of recently described viruses characterized with a tripartite genome and having affinities with *Virgaviridae* (RNA1 and 2) and *Flaviviridae* (RNA3). Using a combination of high-throughput sequencing, datamining and RT-PCR approaches, we demonstrate here that in grapevine samples infected by grapevine-associated jivivirus 1 (GaJV-1) up to 7 additional molecules can be consistently detected with conserved 5′ and 3′ non-coding regions in common with the three previously identified GaJV-1 genomic RNAs. RNA4, RNA5, RNA6, RNA7, RNA8 and RNA10, together with a recombinant RNArec7-8, are all members of a family sharing a previously non recognized conserved protein domain, while RNA9 is part of a distinct family characterized by another conserved motif. Datamining of pecan (*Carya illinoinensis*) public transcriptomic data allowed the identification of two further jiviviruses and the identification of supplementary genomic RNAs with homologies to those of GaJV-1. Taken together, these results reshape our vision of the divided genome of jiviviruses and raise novel questions about the function(s) of the proteins encoded by jiviviruses supplementary RNAs.

## 1. Introduction

The use of high-throughput sequencing (HTS) has revolutionized virus discovery and resulted in an unprecedented flow of novel viral sequences being recognized in association with all environments and organisms, including plants [1,2,3]. Despite these tremendous advances, HTS-based virus discovery is still not a fully straightforward process and a number of pitfalls have been identified [4,5,6]. These include, for example, limitations in the annotation strategies used to identify viral sequences and separate them from host or environmental sequences and the so called “dark matter” that remains refractory to annotation [7]. While a range of new annotation tools have been developed recently [8,9,10,11,12,13], trying to circumvent the limitations imposed to homology-based tools such as BLAST by the incompleteness of databases, the false positive and false negative rates of such tools still remain significant, and substantial uncertainties remain in relation to annotation results. While this may not be a problem in many virome research applications, it has a more obvious impact for virus discovery applications.

Another difficulty that has been identified in the annotation process is that of establishing a link between the fragments of incompletely assembled genomes or between the genomic segments of viruses with divided genomes [4,5,6]. Such cases are frequently observed in metagenomic analyses in which a high viral diversity may be encountered while sequencing depth might be limiting. In the case of viruses with divided genomes, identification of all genomic components might be particularly complicated when they show high variability in number and in encoded protein products between closely related viruses, as has been reported in the case of emaraviruses [14]. As an example, despite having been discovered and sequenced in 2007, two additional genome segments of European mountain ash ringspot-associated virus were only discovered in 2019 [14] and a similar situation has also affected other viruses in this genus. This is also the case with dispensable genomic elements such as the RNA5 found associated with only some isolates of beet necrotic yellow vein virus [15].

When trying to establish links between either fragments of an incompletely assembled viral genome or different genomic molecules of a virus, few clues or strategies are available. As well as homologies or conserved protein motifs with known viruses having well established genomic organizations, key clues can be provided by (i) systematic co-occurrence of fragments/molecules in independent metagenomic datasets that could suggests that they belong to a single agent and (ii) conservation of terminal non-coding sequences, since these show frequently some level of conservation between the different molecules of a divided genome.

The case of jiviviruses illustrates particularly well the difficulties encountered and the strategies that can be employed to try to reconstruct a complete vision of the divided genome of significantly novel viruses. This name was coined in 2020 for unusual viruses showing some affinities with two very distinct groups of viruses, the plant-infected virgaviruses and the insect-infecting Jingmen-like viruses. The first jivivirus sequences were reported in 2017 in a study of the virome of citrus trees affected by the citrus sudden death-associated virus [16]. Interestingly, three contigs of viral nature were identified but not recognized as belonging to the same agent. The contig encoding a protein with homologies to the NS3 of Jingmen virus (*Flaviviridae*) was reported as citrus jingmen-like virus (KY110739) while the two contigs with homologies to the methyltransferase and helicase (ct1, KY110740) and RNA-dependent RNA polymerase (RdRp) (ct2, KY110741) of Virga-like viruses were considered as belonging to a single virus for which the name citrus virga-like virus was proposed [16]. The authors also suggested that these two contigs might be part of a single genomic molecule. Later on, the analysis of virome data of grapevine samples infected by the downy mildew agent, *Plasmopara viticola*, allowed the identification of contigs with clear homologies with the two newly reported citrus viruses [17]. However, the realization that each of the three identified contigs shared conserved 5′ and 3′ non-coding sequences led to the conclusion that they belonged to a single virus having a tri-segmented genome. Given the affinities previously identified with Jingmen and Virga viruses, the name grapevine-associated jivivirus 1 (GaJV-1) was proposed (Jivi = Jingmen-Virga) [17]. This rationale was reinforced by the identification in the same datasets of a second virus with similar genomic organization, which was consequently named grapevine-associated jivivirus 2 (GaJV-2). More recently, the interest of using the conservation of terminal non-coding sequences to identify genome segments of poorly known viruses with divided genomes led to the identification of several potential other jiviviruses from public transcriptome shotgun assembly data (TSA) from several plant species including pecan (*Carya illinoinensis*), *Sarcandra glabra*, *Pinus flexilis* and *Picea glauca* [18]. This study also potentially identified two additional genome segments of GaJV-1 and GaJV-2 and up to three or four additional segments for the Carya jiviviruses and the citrus jivivirus, respectively [18].

Using a combination of HTS analyses, datamining and wet lab experiments, we report here on the analysis and comparison of additional genomic segments of GaJV-1 and Carya jiviviruses and demonstrate that isolates of GaJV-1 may contain a much as nine or ten genomic segments.

## 2. Materials and Methods

### 2.1. Plant Materials

Grapevine samples representing different cultivars were collected in the Bordeaux region (France) in October 2015 and razor blades were used to obtain phloem scrapings that were either immediately processed or stored at −80 °C until used. Old and young leaves were collected from the same vineyard in spring 2019 and immediately used for total RNA extraction.

### 2.2. Double-Stranded RNAs and Total RNAs Purification and HTS

Double-stranded RNAs (dsRNAs) were purified from grapevine phloem scrapings, converted to cDNA and submitted to a random PCR amplification with the addition of terminal multiplex identifier (MID) adaptors as described [19]. PCR products were pooled equimolarly prior to library preparation using the TruSeq Nana kit (Illumina, San Diego, CA, USA). They were finally sequenced in multiplexed format on an Illumina Hiseq3000 platform (2 × 150 bp) outsourced at the GetPlage INRAE platform (Toulouse, France).

Alternatively, total RNAs were purified from phloem scrapings according to [20], converted to cDNA (TrueSeq Stranded Total RNA library kit, Illumina, San Diego, CA, USA) and sequenced at INRAE GeTPlage platform on a Hi-Seq3000 Illumina sequencer (2 × 150 nt paired reads).

### 2.3. HTS Data Analysis and Datamining

Following demultiplexing, reads were trimmed on quality. For reads from total RNA preparations, an additional genome subtraction step was performed using the reference grapevine cv. Pinot noir genome (GCA_000003745.2). A de novo assembly was performed using CLC Genomic workbench (CLC-GW, Qiagen, Hilden, Germany) and BLASTX analysis against the GenBank protein database (nr) used to identify viral contigs [21,22].

Initial datamining for viruses related to GaJV-1 was performed using TBLASTN screening of GenBank plant transcriptome shotgun assembly (TSA) data using the GaJV-1 P1, P2 and P3 proteins as query. Following identification of grapevine and pecan (*Carya illinoinensis*) contigs encoding proteins with similarity to the query proteins, grapevine and pecan sequence reads archives (SRAs) were downloaded, de novo assembled and annotated as described above.

Following the identification of GaJV-1 conserved 5′ and 3′ terminal sequences through multiple nucleotide sequence alignments, the conserved sequences were used as queries in BLASTN searches of locally constructed BLAST databases of contigs from grapevine or pecan SRAs using CLC-GW. Contigs showing 5′ and/or 3′ ends with homologies with the queries were then extended by rounds of mapping of residual reads using CLC-GW to yield genomic scaffolds as complete as possible. The proteins encoded by the identified GaJV-1 additional genomic segments were, in turn, used to screen pecan SRA contigs in TBLASTN searches.

### 2.4. Completion of Identified Genome Segments

The 5′ non-coding region (NCR) of the various genomic segments was completed for the isolate in the i31 grapevine sample using a 5′ RACE kit (Takara Bio Europe/Clontech, Saint-Germain-en-Laye, France) and specific reverse primers designed from the sequence of the reconstructed scaffold for each RNA segment (Appendix A). The 3′ NCRs were completed by polyadenylating the genomic segments (Ambion/ThermoFisher Scientific, Illkirch, France) and performing polyA-anchored RT-PCRs using LD-polyT as a reverse primer and a specific forward internal primer designed from the sequence of the reconstructed scaffold for each RNA segment (Appendix A) as described [23]. An internal gap in the i31 RNA5 segment was determined by direct sequencing of a RT-PCR fragment obtained using internal primers (CVJ-i31-RNA5-trou-F/CVJ-i31-RNA5-trou-R) (Appendix A).

### 2.5. Total Nucleic Acids Extraction, Virus Detection by RT-PCR, Fungal Detection by Generalist PCR Targeting Ribosomal Internal Transcribed Spacers (ITS)

Total nucleic acids were extracted from grapevine leaves or phloem scrapings using the Spectrum™ Plant Total RNA Kit (Sigma-Aldrich, St. Louis, MI, USA) with 3% PVP40K according to the manufacturer’s recommendations. In some cases, in order to remove possible epiphytic fungi, prior to RNA extraction, leaves were surface-washed with sterile water and surface-disinfected by soaking for ten minutes with occasional agitation in a fresh 5% sodium hypochlorite solution. This was followed by six rinses in sterile water before air-drying on filter paper and proceeding with RNA extraction.

Complementary DNA was synthetized with the Thermo Scientific Maxima H Minus reverse transcriptase and a mix of pdN_6_ and dT18 primers using the manufacturer’s instructions. Forward and reverse primers were selected for each RNA segment so as to amplify PCR fragments of different length and thus allow to detect several segments in a single PCR reaction (Appendix A). PCR amplifications were performed using the Thermo Scientific DreamTaq DNA polymerase according to the manufacturer’s instructions with the following cycling scheme: 5 min at 95 °C, 40 cycles at 94 °C for 30 s, Tm for 30 s, 72 °C for 60 s and a final 72 °C for 10 min. The annealing temperatures used (Tm) are indicated in Appendix A. PCR products were analysed by electrophoresis on non-denaturing 1% agarose gels and directly sequenced to verify amplification specificity.

Primers ITS1-F 5′-CTTGGTCATTTAGAGGAAGTAA-3′ and ITS2 5′ GCTGCGTTCTTCATCGATGC-3′, described in [24,25] and targeting the internal transcribed spacer region of the fungal ribosomal RNA operon were used in RT-PCR assays to detect the presence of fungi. Amplification conditions were similar as described above, with the following cycling scheme: 5 min at 95 °C, 40 cycles at 94 °C for 30 s, 55 °C for 30 s, 72 °C for 30 s and a final step at 72 °C for 10 min.

### 2.6. Sequence Comparisons, Conserved Protein Domains Searches, Recombination Analysis, Phylogenetic Analyses

Multiple sequence alignments were performed with ClustalW [26] implemented in MEGA7 [27]. The presence of conserved protein motifs in the various proteins encoded by the genomic RNAs of GaJV-1 and CaJV-1 and CaJV-2 was investigated using NCBI conserved domain search tool (CD-search https://www.ncbi.nlm.nih.gov/Structure/cdd/wrpsb.cgi, last accessed on 15 September 2022), HMMER [13] or multiple alignments of the various proteins. Recombination analysis was performed using RDP4 [28] with default parameters and Bonferonni correction for the statistical evaluation of the significance of identified events. Phylogenetic trees were reconstructed in MEGA7 using the neighbor-joining method and strict amino acid distances and branch validity was evaluated by bootstrap analysis (1000 replicates).

## 3. Results

### 3.1. Identification of Genomic Segments of Grapevine Associated Jivivirus 1 in French Grapevines

As part of a project investigating the virome of the grapevine cultivars and clones of the Bordeaux area (France), HTS RNASeq analysis was performed on total RNAs extracted from the leaves or from phloem scrapping of grapevines collected in September 2015. The datasets were analyzed by de novo assembly using CLC Genomics Workbench (Qiagen, Aarhus, Denmark) followed by contigs annotation using BLASTN and BLASTX analysis against the GenBank database. This allowed the identification of several grapevine samples which yielded contigs with sequence similarity to citrus virga-like and citrus jingmen-like viruses, respectively, and which are now known to represent the genomic RNAs of GaJV-1 (Table 1). However, the numbers of reads involved in the identified contigs were relatively low (514 to 3401 reads, depending on the sample) and represented only a very minor fraction of total reads (0.003 to 0.024%). This prompted an effort to generate more sequence information for this poorly known virus. Double-stranded RNAs (dsRNAs) were purified from phloem scrapings and analyzed by HTS following their conversion to cDNA by a random amplification procedure [19]. The four grapevine samples analyzed in this fashion showed evidence of GaJV-1 infection, with two of them, i31 and i33, showing the most reads for GaJV-1 genomic RNAs (respectively, 1.6 and 1.5% of total reads). Further analysis was therefore focused on the contigs assembled for the i31 sample, allowing the identification, as reported independently [17,18], of homologies in the 5′ and 3′ non-coding regions (NCRs) of the contigs for the three GaJV-1 genomic RNAs. These conserved 5′ and 3′ NCRs were then used in BLASTN searches of all the contigs assembled from these two samples. This allowed the identification of further contigs characterized by the presence of one or both of these terminal conserved regions but having different internal sequences. Six such contigs were identified in both datasets. Since the sharing of the conserved terminal regions suggested that the identified molecules could be additional genomic segments of GaJV-1, they were correspondingly named RNA4 to RNA9, with RNA5 only detected as a small contigs of few reads in sample i31 and the other genomic RNAs having coverages or between 158x (RNA4) and 790x (RNA9) (Table 1).

The ends of all identified GaJV-1 molecules were completed on the i31 sample, using 5′ RACE and, following polyadenylation, 3′ LD-PCR [23]. Given the partial contig obtained for the RNA5, an internal gap was closed by a targeted PCR amplification. The complete sequences of all identified genomic RNAs from the i31 isolate have been deposited in GenBank under accession numbers OP428756-64. Figure 1 presents multiple alignments for the 5′ and 3′ NCRs of all molecules, together with the sequences for the RNA1, RNA2 and RNA3 of the reference isolate (MN520745, MN520746, MN520747) [17]. As can be seen, sequence conservation is more extensive for the 5′ NCR, including a 45 nt region in which only four positions are not fully conserved between all genome segments. The alignment of the 5′ NCR also indicates that a few nucleotides are missing from the ends of the reference isolate (respectively, 13, 33 and 10 nt for RNA1, RNA2 and RNA3).

The 3′ NCR is less conserved but still contains several regions of significant conservation, in particular in the region of nt 2-31 in the alignment, with only nine not fully conserved positions. Interestingly, the 5′ NCR most conserved region is A and, to a lesser extent, C rich, while the 3′ NCR most conserved regions are G and T rich.

### 3.2. Datamining Identifies Two Additional Genome Segments in Italian Isolates of GaJV-1

In order to see if additional genomic segments might be present in other GaJV-1 isolates, datamining was then performed on grapevine RNASeq transcriptomic data publicly available as sequence reads archives (SRA). A first screen identified five deep datasets from different grapevine cultivars from Italy deposited by the Research and Innovation Centre, Fondazione Edmund Mach. These correspond to the Polöskei Muskotaly (ERX1002380), Moscato Rosa (ERX1002659), Lambrusco (ERX1002378), Teroldego (ERX1002620) and Alicante Bouschet (ERX1002377) cultivars, and had GaJV-1 reads representing between 0.001% and 0.03% of total reads (Table 1). A BLASTN search of the contigs assembled from these datasets using the conserved 5′ and 3′ NCRs allowed us to identify from the Teroldego dataset two further molecules sharing these genomic ends but having different internal sequences. One of these molecules turned out to be a recombinant between the already identified RNA6 and RNA7 and was therefore named RNArec7-6 (see below), while the second, which had not been previously identified, was named RNA10. The sequences of these two additional genome segments have been deposited in GenBank under the BK061938-39 accession numbers. The average coverage for the identified contigs were respectively 36x (RNArec7-6) and 25x (RNA10). In addition, two variants could be assembled from the Teroldego dataset for both RNA3 and RNA4. For the RNA3, the assembly for one of the variants (3-1) still contained a small internal 118 nt gap, while the contig for the second variant (3-2) was shorter (1.4 kb) and missed significant 5′ and 3′ sequences. The same situation was found for RNA4, with the 4-2 variant (1.6 kb) missing significant 5′ and 3′ sequences. The 3-1 and 3-2, and 4-1 and 4-3 variants showed, respectively, 10.0% and 9.9% divergence with each other. Since all these molecules were identified by datamining, no effort was performed to extend them and determine their extreme 5′ and 3′ ends. All GaJV-1 genomic RNA sequences identified have been deposited in GenBank under OP428756 to OP428764 (i31 isolate) and BK061930 to BK061939 (Teroldego isolate) accession numbers.

### 3.3. Reads Mapping Provides Evidence for Limited Presence/Absence Variability for the Various Genomic Segments Identified in French and Italian Isolates of GaJV-1

In a first attempt to investigate presence/absence variability for the various genomic segments, quality trimmed reads for the dsRNA and total RNA French datasets and for the RNASeq Italian datasets were mapped against the nine full-length molecules from the French i31 isolate and against the two additional molecules identified from the Italian Teroldego sample. Mapping parameters were adjusted to try to avoid cross-mapping of reads to different reference sequences, in particular in the conserved terminal regions.

The results show only limited variability in the presence/absence of the various genomic molecules (Table 1). In particular, with the exception of the Italian Teroldego dataset, RNA1 to RNA4, RNA6, RNA7, RNA8 and RNA9 were detected in all analyzed samples. RNA10 was detected in all five Italian datasets but in only three of the eleven French samples, with very few reads in two of them. Similarly, RNA5 was not detected in three French samples and only weakly in three of them. Lastly, the Teroldego dataset was unique in missing RNA5, RNA6 and RNA7 but conversely containing the recombinant RNArec7-6 (Table 1). Interestingly, average coverage of this recombinant molecule (35.6x) was of the same order to that observed for the other additional genome segments identified here (respectively, 30.5x (RNA4), 18.4x (RNA8) and 25.3x (RNA10)), suggesting they all are able to accumulate to similar levels and that RNArec6-7 is not merely an artifact of low fitness. Taken together, these results show that the various RNAs analyzed show a strong tendency to be identified as a group, reinforcing the notion that they represent parts of the GaJV-1 genome.

### 3.4. RT-PCR Detection of the Various Genomic Segments from Vineyard Samples

In order to validate the presence of RNA1 to RNA9 in grapevine samples, primer pairs specific for each RNA were designed (Appendix A). The size of the amplicons was selected so as to allow the development of multiplexed assays that would allow the simultaneous detection of all genomic RNAs (Appendix A). As expected, all nine RNAs could be amplified from RNA extracts that had been used for HTS sequencing. Resampling of grapevine leaves in spring 2019 in the same collection vineyard in which the original samples had been collected in 2015 allowed us to identify grapevines from which all GaJV1 RNAs could be detected using both young or old leaves (Figure 2).

Surface sterilization of the leaves using sodium hypochlorite prior to total nucleic acids extraction did not significantly affect the obtained amplification signals. On the other hand, this surface sterilization treatment almost completely abolished the fungal amplification signal obtained using polyvalent primers ITS1F and ITS2 targeting the internal transcribed spacer of fungal ribosomal RNA. Taken together, these results suggest that GaJV-1 is not a mere surface contaminant of the tested leaves.

### 3.5. Genome Organization of GaJV-1

For each of the identified genomic RNAs, a single open reading frame (ORF) taking up most of the molecule was identified. The overall genomic organization of GaJV-1 is shown in Figure 3 while the precise characteristics of each genomic RNA are given in Table 2.

The 5′ NCRs are relatively homogenous in length, ranging from 89 nt (RNA1 and RNA3) to 152 nt (RNA8). Sequence comparisons suggest that for Teroldego isolate RNA10 and RNArec7-6, respectively, 12 and 16 nt are missing at the 5 ends of the contigs and, respectively, 21 and 41 nt at their 3′ end. This would make the 5′ NCR of RNA10 the longest of all at 174 nt. The 3′ NCRs are more variable in length, ranging from 70 nt (RNA3) up to 456 nt for RNA8. RNA1 to RNA4 encode proteins of, respectively, 145.8, 114.8, 72.2 and 60.7 kDa, while the proteins encoded by all other RNAs are relatively homogenous in size, ranging from 41.7 kDa (RNA7) to 52.7 kDa for the protein encoded by the recombinant RNArec7-6.

The existence of a recombination event linking RNA6, RNA7 and RNArec7-6 was confirmed by a RDP4 analysis [28], with six of seven programs detecting the event with a Bonferonni corrected probability of 1.5 × 10^−60^ and a breakpoint identified in the 835-899 region of the rec7-6 molecule, a small region in which RNArec7-6 diverges extensively from both RNA6 and RNA7. In the 5′ upstream part of the molecule, RNArec7-6 shows 45.7% nt divergence with the i31 RNA6 sequence but only 6.0% nt divergence with the RNA7 one. The situation is reversed in the 3′ part of the molecules with 6.3% and 38.6% nt divergence with i31 RNA6 and RNA7, respectively.

For the RNA1, the i31 isolate sequence shows 88.9% nt identity with that of the Italian reference isolate (MN520745), while the sequence assembled from the Teroldego dataset is 99.8% identical. For the RNA2, the corresponding values are 90.7% and 86.8%, with the i31 and Teroldego sequences themselves being 90.6% identical. Lastly, for the RNA3, the i31 sequence is 91.6% identical with the reference isolate (MN520747), while the Teroldego 3-1 and 3-2 molecules are, respectively, 97.8% and 92.6%, identical with that same reference isolate.

### 3.6. Additional Genome Segments in Two Other Jiviviruses Identified in Pecan (Carya illinoinensis)

A TBLASTN screening of GenBank plant transcriptome shotgun assembly (TSA) data using the GaJV-1 P1, P2 and P3 proteins as query-identified homologous sequences in pecan (*Carya illinoinensis*) transcriptomic data generated in China [29]. All SRAs corresponding to the TSA were retrieved, separately assembled and screened by BLAST for the presence of viral sequences homologous to GaJV-1. Such sequences were identified in two out of twelve pecan SRAs (SRX3209418 and SRX3209419) and, when needed, identified contigs were scaffolded and extended by rounds of mapping of residual reads. Interestingly, sequence comparisons suggested that the viruses identified in these two SRAs might represent different agents (see below), which were therefore named Carya-associated jivivirus 1 (CaJV-1, from SRX3209419) and Carya-associated jivivirus 2 (CaJV-2, from SRX3209418). Comparison of the 5′ and 3′ NCRs for the identified RNA1, RNA2 and RNA3 for each virus allowed to identify regions conserved between the different genomic RNAs. These conserved regions were then used as queries in BLASTN searches of all assembled contigs. Searches were also performed by TBLASTX using the sequences of GaJV-1 additional RNAs as queries. These various searches allowed us to identify one additional contig with homologies to GaJV-1 RNA9 among the SRX3209418 contigs, which was correspondingly named CaJV-2 RNA9. Among the SRX3209419 contigs, five additional contigs were identified, one with homologies to GaJV-1 RNA9 and four sharing a ca. 130 aa domain with homologies to GaJV-1 RNA4 to RNA8, but otherwise highly variable and correspondingly named CaJV-1 RNAs A, B, C and D. Given that these various pecan contigs were obtained by datamining, no specific efforts could be made to determine their precise 5′ and 3′ ends. All CaJV-1 and CaJV-2 identified contigs have been deposited in GenBank under accession numbers BK061940 to BK061947 (CaJV-1) and BK061948 to BK061951 (CaJV-2). The genomic organization of all identified CaJV-1 and -2 RNAs identified is the same as for GaJV-1, with a single ORF representing most of the molecule length as shown in Table 3. Given that the 5′ and 3′ NCRs are truncated, it proved more difficult to identify conserved regions between the different genome segments, in particular for CaJV-2 for which only a CTTACTTG(n)AT(nn)C(n)ACCTAAA(n)TAAAC(n)A could be identified, showing some limited homology with the longer conserved sequences shared by the 5′ NCRs of GaJV-1 or of CaJV-1 (Figure 1 and Appendix A, respectively).

When complete ORFs are present in the identified contigs, the length of the encoded proteins for CaJV-1 and -2 are comparable to those of GaJV-1 and significant identity levels are observed. Proteins P1, P2 and P3 show, respectively, 48.3%, 62.7% and 54.8% identity with those of GaJV-1 and have closely matching sizes (Table 3). For its part, P9 shows 53.2% identity while being about 10 kDa larger (Table 3). In the case of CaJV-1, the situation is very similar, even if the exact size of the P1 ORF cannot be ascertained because of an internal gap in the scaffold. Identity level with the corresponding proteins of GaJV-1 are however comparable with 45.2%, 62.5%, 54.1% and 51.6% aa identity with the GaJV-1 proteins for P1, P2, P3 and P9, respectively. The corresponding values when comparing the proteins of CaJV-1 with those of CaJV-2 are in the same range, being, respectively, 57.8%, 69.3%, 62% and 53.5% (Table 3), indicating that the two viruses are slightly more closely related than they are to GaJV-1.

The situation is very different for RNA A, B, C and D since they are both variable in length (between 41.6 and 64.5 kDa) and highly variable in sequence. Indeed, they show amino acid identities ranging only from 15.6% (RNA A and RNA B) to 38.7% (RNA A and RNA C) and only between 10.2% and 19.9% identity with proteins P4 to P8 of GaJV-1, with which they share a conserved protein domain of about 130 amino acids (Figure 3).

### 3.7. Conserved Protein Motifs in the Proteins of GaJV-1 and CaJV-1

The presence of conserved protein motifs in the various proteins encoded by the genomic RNAs of GaJV-1 and CaJV-1 and CaJV-2 was investigated using NCBI conserved domain search tool (CD-search https://www.ncbi.nlm.nih.gov/Structure/cdd/wrpsb.cgi, accessed on 15 September 2022), HMMER [13,30] or multiple alignments of the various proteins. The expected methyl-transferase, helicases and RdRp conserved domains were identified in proteins P1, P2 and P3 [17], Figure 3. In the P4 protein of GaJV-1, a nucleoside ribosyltransferase domain (pfam15891.8) was identified and BLASTP identified weak homology with bacterial and fungal proteins containing a similar domain (best e-value 8.0 × 10^−7^). Very weak homology with bacterial proteins with a vapC domain (COG1487, corresponding to nucleic acid-binding proteins containing a PIN domain, Figure 3) was identified by BLASTP for GaJV-1 P8 protein (best e-value 1.0 × 10^−6^). No known conserved domain or homology with a protein in the GenBank database could be identified for proteins P5, P6, P7, Prec7-6, P8 and P10 of GaJV-1 or for proteins PA, PB, PC or PD of CaJV-1. However, a multiple alignment revealed the existence of a conserved domain shared between all of these proteins, as well as with GaJV-1 P4 and spanning about 130 amino acids, of which 10 were absolutely conserved (Figure 4).

Similarly, no known conserved domains were identified by HMMER or CD-search in the P9 proteins of GaJV-1, CaJV-1 or CaJV-2. However, two shared domains were identified in BLASTP analyses or in multiple alignments (Figure 3). The first one is a ca. 100 aa region in the central part of the P9 proteins (aa 67-171 of the GaJV-1 protein) which is also observed near the N-terminus of GaJV-2 P3 protein (aa 33-138, Appendix A). The second is a highly conserved C-terminal region shared by the P9 proteins of the three viruses (Figure 3 and Appendix A).

### 3.8. Phylogenetic Affinities of the Analyzed Viruses

Phylogenetic trees were reconstructed using the sequences of the P1, P2 or P3 proteins of the GaJV-1 and CaJV-1 or -2 reported here, and those of the closest viruses identified by BLASTP analysis in GenBank. These references include citrus virga-like and citrus jingmen-like viruses [16], grapevine-associated jiviviruses 1 and 2 [17] and some newly reported agents, such as mastic virus Y (MVY) identified in *Pistacia lentiscus* (MT334608-10), sisal-associated virgavirus B from the virome of sisal plants [30,31], soybean thrips Jivi-like viruses 1 and 2 from suction trap collected soybean thrips (*Neohydatothrips variabilis*) [32] and *Aspergillus lentulus* jivivirus 1 from a culture of the human fungal pathogen *Aspergillus lentulus* [33]. The tree for the RdRp encoding P2 protein is shown in Figure 5 while those for the P1 and P3 proteins are, respectively, shown in Figure 6 and Appendix A.

Although the three trees show some differences in the branching order of the reference viruses used, they provide a similar message when it comes to the GaJV-1 and CaJV-1 or -2 sequences reported here. The two GaJV-1 sequences are extremely closely related to the reference GaJV-1 sequence, as was expected from the percentages of identity in pairwise comparisons. This result confirms that all analyzed isolates indeed belong to GaJV-1 and show only limited divergence between French and Italian isolates. For each protein, the two CaJV viruses are more closely related to each other than to any other virus and, together with GaJV-1 and MVY, for a well separated cluster, supported by very strong bootstrap values. In the P2 tree, soybean thrips jivi-like virus 2 also clusters in that group but it is distinctly divergent and clusters away in the P1 tree.

## 4. Discussion

The first jivivirus was initially identified as two distinct agents during the analysis of the virome of citrus plants [16], pointing to its unusual phylogenetic affinities with widely different virus families, the *Virgaviridae* and the *Flaviviridae*. A later study in grapevine identified homologous genomic segments to the citrus viruses, but the discovery that the two virga-like segments and the flavi-like segment were systematically associated in the virome of different grapevines, and shared conserved 5′ and 3′ NCRs led to the conclusion that they represent the genome of a tripartite virus, GaJV-1 [17]. This conclusion was further reinforced by the identification of a second tripartite virus with the same genomic organization and affinities, GaJV-2 [17]. The results presented here, investigating systematic or near systematic association of molecules in planta and in the viromes of different grapevines, and 5′ and 3′ NCRs conservation between genome segments, indicate that the genome of GaJV-1 is much more complex than previously understood and may contain up to seven additional genomic segments.

The notion that the genomes of jiviviruses may contain additional genome segments of the three previously identified was recently proposed on the basis of a datamining study investigating conserved genome ends [18]. However, this study only identified one additional segment for GaJV-1 and two additional segments in two viruses identified from pecan datasets and that are presumably identical to the CaJV-1 and CaJV-2 viruses reported here. The highest number of additional segments identified is three for the citrus virus [18].

The availability of grapevine HTS datasets generated from highly purified dsRNA preparations has been critical for the identification of the additional GaJV-1 genome segments. Indeed, while GaJV-1 was only 0.001 to 0.03% of total reads in datasets generated from total RNAs, it reached up to 2.67% in the dsRNA datasets or a nearly 100-fold higher representation (Table 1). This in turn provided high coverage, allowing the assembly of nearly complete molecules, including their NCRs (Table 2) which was critical for their recognition as GaJV-1 genomic segments. Indeed, since they lack conserved protein motifs identified as being associated with viruses or homologies with other viral sequences in GenBank, these molecules would have been extremely difficult to recognize as viral in nature and would have remained unidentified and un-annotated. As discussed in [18], the systematic search for molecules sharing conserved 5′ and 3′ ends clearly represents an under-utilized strategy for the identification of genomic segments of viruses with divided genomes.

In the case of the pecan viruses, the identification of the additional RNAs relied both on conserved genomic ends and on homologies with the newly identified GaJV-1 genomic segments, suggesting that similar molecules could likely be identified for other jiviviruses [18]. The reads mapping results from a range of grapevine samples indicate that the whole complement of seven additional RNAs is not systematically present in GaJV-1 isolates, suggesting a level of flexibility and the idea that at least some of the identified RNAs might be dispensable (Table 1). However, RNA4, RNA6, RNA7, RNA8 and RNA9 were systematically present with a single exception: when absent from the Teroldego isolate, RNA6 and RNA7 were replaced by the RNArec7-6 recombinant. On the other hand, RNA5 was not identified in four isolates and RNA10, present in all five Italian datasets, was only detected in 3/11 French samples (Table 1). The function (s) of these additional molecules is unknown and their lack of homology with any other proteins in GenBank does not help in proposing hypotheses. In this way, these additional genomic segments are similar to the additional segments identified in variable number in the genome of different emaraviruses [14].

A remarkable pattern has however emerged when it comes to RNA4 to RNA8 and RNA10 of GaJV-1 or RNA A to RNA D of CaJV-1 since while being extremely variable in both length and sequence, the proteins they encode all share a conserved motif that had not been previously identified. This suggests a scenario in which these proteins could serve as adapters between a common partner and different cellular factors recognized by each viral protein, similar to the families of effectors expressed by fungal or bacterial pathogens to target host defense or sensing machineries, in particular through the ubiquitin-proteasome pathway [34,35,36]. Interestingly, datamining of the *Aspergillus lentulus* dataset from which AlJV1 had been identified (ERR7929648, [33]) allowed the identification, as well as the three known genomic RNAs of AlJV1, of two further contigs encoding proteins showing, respectively, homologies with the proteins P4 (e-value 7.4 × 10^−8^) and P9 (e-value 5.9 × 10^−43^) of GaJV-1. The protein with distant homology to GaJV-1 P4 shares the conserved domain identified in proteins P4, P5, P6, P7, P8 and P10 of GaJV-1 and shown in Figure 4 and also possesses the conserved nucleoside ribosyltransferase domain (pfam15891) present in GaJV-1 P4. The P9 protein encoded by RNA9 of both GaJV-1 and CaJV-1 and -2 is the only protein that does not belong to the family. Interestingly, while TBLASTN search of plant TSA using proteins of the other family yielded few hits with generally low e-values, with notable exception being hits in the mango TSA (*Mangifera indica*, e^−13^ to e^−24^ e-values), multiple hits were identified using P9 proteins as queries. A 920 nt almond contig (*Prunus dulcis*, GJSC01043913) was identified with 96.8% nt identity with GaJV-1 RNA9 and a scaffold assembled from two giant redbud contigs (*Cercis gigantea*, GAOK01008846 and GAOK01008847, total length 939 nt) showed 99.2% nt identity with CaJV-1 RNA9. However, more importantly, contigs showing strong homology with the conserved C-terminal domain of P9 proteins were identified in the TSA from a variety of plants including *Bletilla striata*, *Persea americana*, *Viola orientalis*, *V. albida, Ilex paraguanensis*, *Iris domestica*, *Tectona grandis*, *Lumnitzera littorea*, *Selaginella lepidophylla*, *Mangifera indica*, *Cenostigma pyramidale*, *Aquilaria malacensis*, *Vriesea carinata*, *Gleditsia sinensis*, *Bambusa oldhamii*, *Sarcandra glabra*, *Ardisia crenata*, *Phalenopsis aphrodite*, *Gingko biloba* and *Juniperus ashei* (Appendix A). For several species, the identified contigs contained A rich repeats, typical of the 5′ NCR of GaJV-1 or CaJV-1 (Figure 1 and Appendix A). Taken together, these results suggest that viruses with genomic RNAs homologous to GaJV-1 RNA9 might be associated with a range of plants including dicots, monocots, gymnosperms and even primitive plants such as the lycophyte *Selaginella lepidophylla*. In contrast, no contigs encoding proteins with homologies to P9s were identified in fungal TSAs while a few contigs were identified in insect TSAs, in particular *Diaphorina citri* (GDFN01015706.1, e-value 4.0 × 10^−89^) and *Hedychridium coriaceum* (GBQW01009463.1, e-value 5.0 × 10^−69^). However, as indicated above a protein with homologies with GaJV-1 P9 was identified from the assembly of the *Aspergillus lentulus* dataset from which the AlJV1 has been identified (ERR7929648, [33]).

Although these preliminary results suggest that viruses with P9-like proteins are more frequently associated with plants than with fungi or insects, the question of the host of GaJV-1 and of comparable viruses remains an open one. The viruses of the cluster involving GaJV-1, CaJV-1 and -2 and MVY were all identified from plant samples. However, related viruses were identified from fungal culture (AlJV-1, [31]) or from insects (STJlV-1 and -2, [32]). In particular, the *Aspergillus lentulus* data from which AlJV1 was identified correspond to the analysis of a pure fungal culture supernatant and showed deep coverage (601x to 1985x) of the five identified contigs encoding proteins with homologies to those of GaJV-1, strongly supporting the notion that JaJV1 is indeed a fungal virus, and that such may therefore also be the case for other jiviviruses. The RT-PCR detection experiments from surface disinfected leaves, which showed a loss or severe reduction of the signal from fungal ITS sequences but no impact on GaJV-1 detection, suggest that this virus was not detected as a mere surface contaminant and so does its HTS detection from nucleic acids extracted from phloem scrapings. Taken together, these results suggest that GaJV-1 is present internally in grapevine tissues but do not demonstrate that grapevine is a host of GaJV-1, as it could also be infecting a fungal endophyte of grapevine. Further experimental efforts are now needed to better understand its biology and that of other related jiviviruses, and to begin the unravel the contribution to viral infection of the novel genomic components identified here.

## Figures and Tables

**Figure 1 viruses-15-00039-f001:**
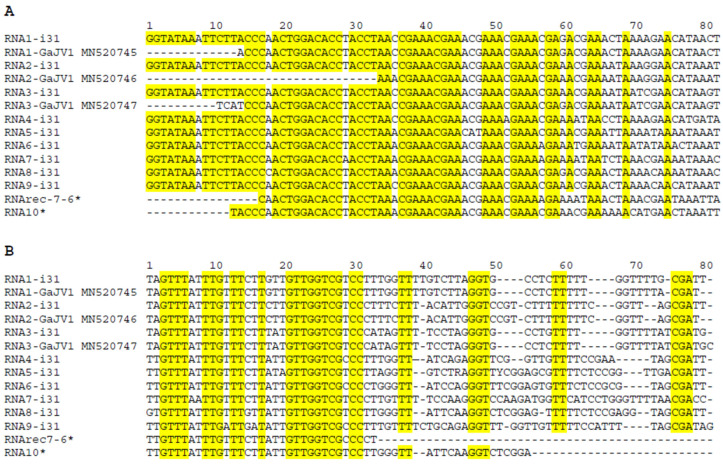
Multiple nucleotide sequence alignments of (**A**) the 5′ non coding region (NCR) and (**B**) the 3′ NCR of the various genomic segments identified for grapevine-associated jivivirus 1 (GaJV-1). The i31 isolate sequences determined here are shown, together with the RNArec 7-6 and RNA10 from the Teroldego isolate (marked by an asterisk) and with the RNA1, RNA2 and RNA3 sequences for the reference isolate (MN520745, MN520746, MN520747). Residues conserved in all RNA segments are highlighted in yellow.

**Figure 2 viruses-15-00039-f002:**
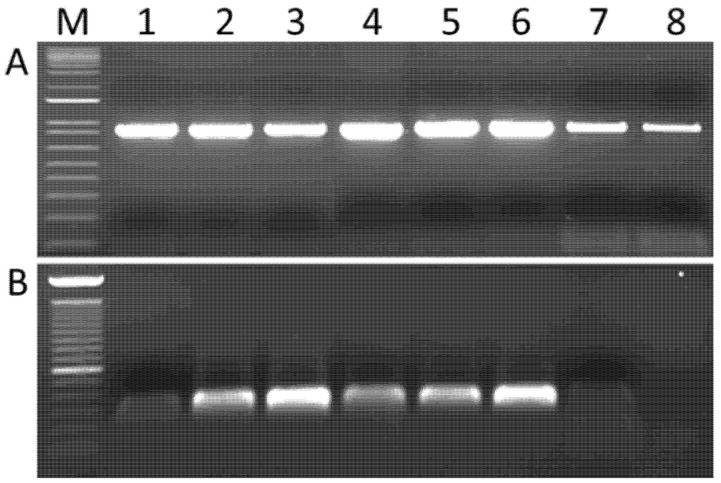
Detection of (**A**) GaJV-1 and of (**B**) fungal ITS sequences from total nucleic acids from grapevine leaves. (**A**) RT-PCR amplification of GaJV-1 RNA5 using the GaJV-1 RNA5F/RNA5R primer pair. (**B**) PCR amplification of fungal ITS sequences using the ITS1F-ITS2 primer pair. M, molecular weight markers ladders. Lanes 1 to 3, amplification from young grapevine leaves. Lanes 4 to 6, amplification from old grapevine leaves. Lanes 7 and 8, amplification from young grapevine leaves following sodium hypochlorite surface disinfection. The gel pictures have been spliced to remove additional samples.

**Figure 3 viruses-15-00039-f003:**
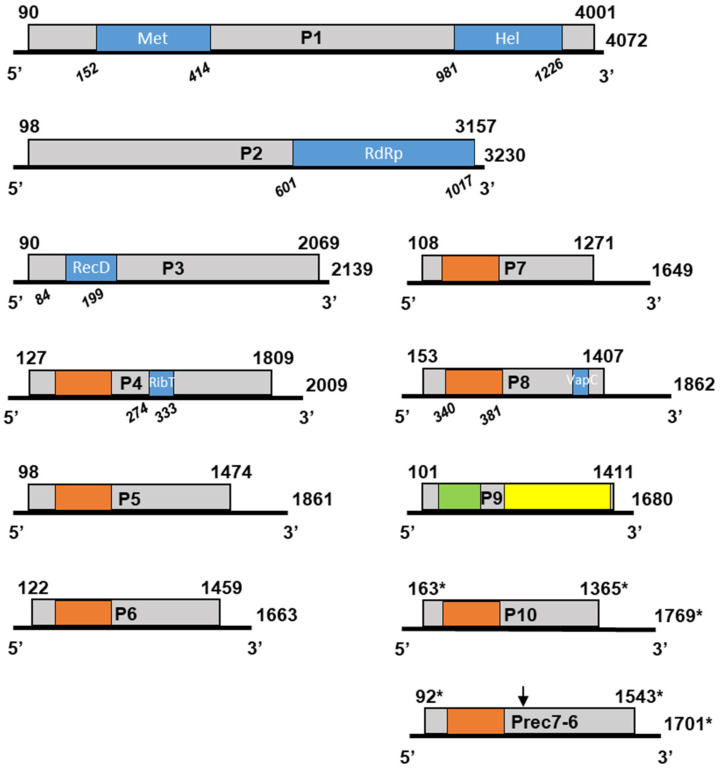
Schematic genome organization of GaJV-1. Boxes represent the open reading frames (ORF) encoded on each genomic RNA. Nucleotide positions of ORFs are indicated above, while molecule length is indicated on the right. Conserved protein motifs are indicated by blue boxes, with amino acid positions are indicated in italic below. The motifs identified are Met: viral methyl-transferase family (cl03298), Hel: viral helicase 1 family (pfam01443), RdRp: RNA-dependent RNA polymerase 2 family (pfam00978), RecD: RecD family helicase (cl37724), RibT: nucleoside ribosyltransferase domain (pfam15891.8), VapC: virulence-associated protein C domain (COG1487). A ca. 130 aa region shared between the P4, P5, P6, P7, P8, P10 and Rec7-6 proteins is indicated by orange boxes. A ca. 100 aa region shared between GaJV-2 P3 and the P9 of Carya jiviviruses 1 and 2 is indicated in green, while a ca. 250 aa region shared between the P9 of Carya jiviviruses 1 and 2 is indicated in yellow. Nucleotide positions for RNA10 and RNArec7-6 are marked by an asterisk to indicate that 5′ and 3′ ends have not been determined.

**Figure 4 viruses-15-00039-f004:**
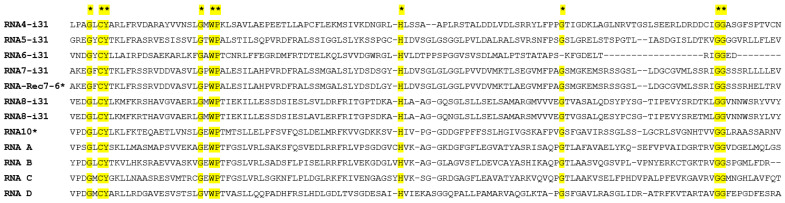
Multiple alignment of the conserved domains identified in selected GaJV-1 and CaJV-1 proteins. Fully conserved amino acids are highlighted in yellow and marked by an asterisk above the alignment.

**Figure 5 viruses-15-00039-f005:**
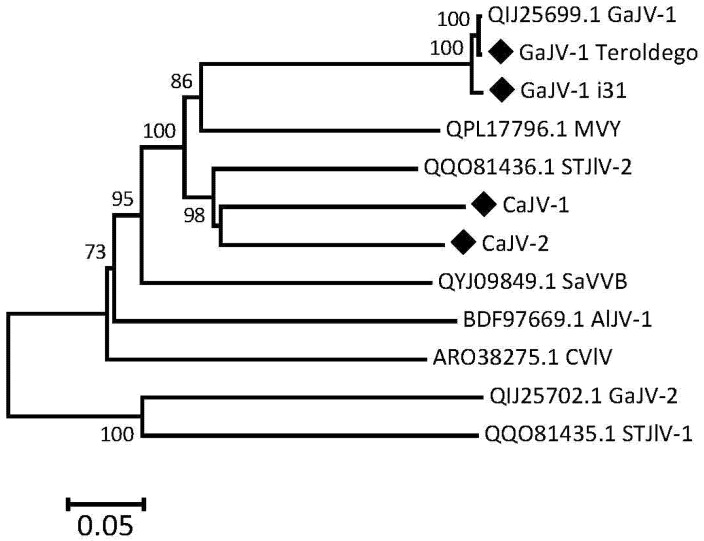
Phylogenetic tree reconstructed from the sequences of the P2 proteins of GaJV-1, CaJV-1 and selected related agents. The tree was reconstructed using the neighbor-joining method and strict amino acid distances. Branch validity was evaluated by bootstrap analysis (1000 replicates) and only bootstrap values > 70% are shown. Sequences determined here are marked by a black diamond. The scale bar represents 5% aa divergence. GaJV-1 (or -2): grapevine-associated jivivirus 1 (or -2); CaJV-1 (or -2): Carya-associated jivivirus 1 (or -2); CVlV: citrus virga-like virus; MVY; mastic virus Y; SaVVB: sisal-associated virgavirus B; AlJV-1: *Aspergillus lentulus* jivivirus 1; STJlV-1 (or -2): soybean thrips Jivi-like virus 1 (or -2).

**Figure 6 viruses-15-00039-f006:**
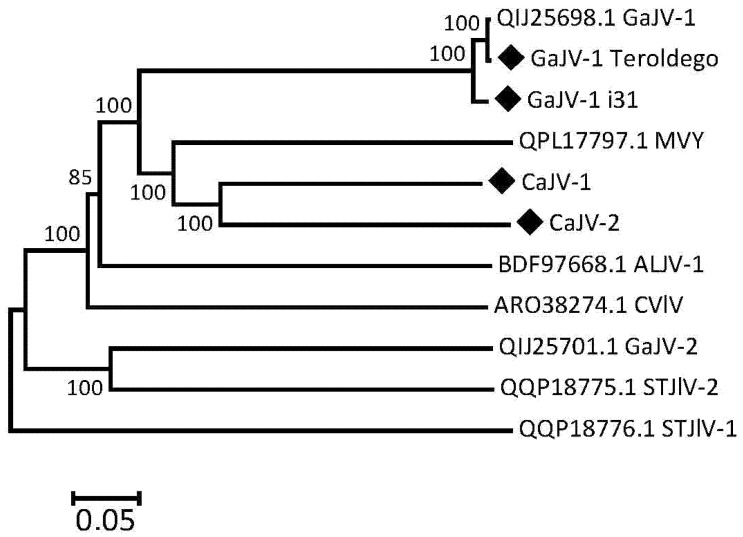
Phylogenetic tree reconstructed from the sequences of the P1 proteins of GaJV-1, CaJV-1 and selected related agents. The tree was reconstructed using the neighbor-joining method and strict amino acid distances. Branch validity was evaluated by bootstrap analysis (1000 replicates) and only bootstrap values > 70% are shown. Sequences determined here are marked by a black diamond. The scale bar represents 5% aa divergence. GaJV-1 (or -2): grapevine-associated jivivirus 1 (or -2); CaJV-1 (or -2): Carya-associated jivivirus 1 (or -2); CVlV: citrus virga-like virus; MVY; mastic virus Y; AlJV-1: *Aspergillus lentulus* jivivirus 1; STJlV-1 (or -2): soybean thrips Jivi-like virus 1 (or -2).

**Table 1 viruses-15-00039-t001:** Mapping of reads from various grapevine HTS datasets on the GaJV-1 genomic RNAs.

Code/SRA *	Cultivar	Trimmed Reads	^%^ Mapped Reads ^$^	RNA1 ^%^	RNA2 ^%^	RNA3 ^%^	RNA4 ^%^	RNA5 ^%^	RNA6 ^%^	RNA7 ^%^	RNA Rec7-6 ^#^	RNA8 ^%^	RNA9 ^%^	RNA10 ^#^
i9 (ds)	Petit Verdot	1,928,756	0.31%	2446	1215	471	361	3	110	351	-	527	524	-
i30 (ds)	Cabernet Sauvignon	1,909,912	1.79%	12,940	5031	3313	1238	423	879	2270	-	5434	2621	-
i31 (ds)	Merlot	3,345,630	2.67%	32,957	1,3072	7678	2560	7	2690	8354	-	11,302	10,747	8
i33 (ds)	Cabernet franc	1,435,868	2.23%	12,678	5895	2915	1166	456	901	861	-	3530	3684	1
i10-2	Semillon	12,385,467	0.02%	746	498	123	105	-	78	85	-	53	443	-
i7-3	Semillon	12,852,574	0.01%	603	351	171	176	-	25	147	-	87	240	-
i11-2	Merlot	14,094,852	0.04%	1759	1093	549	170	62	465	342	-	289	882	96
i14-3	Merlot	20,767,361	0.01%	445	301	188	77	-	52	111	-	64	239	-
i6-3	Cabernet Sauvignon	22,610,049	0.01%	575	492	301	37	92	124	47	-	89	533	-
i25-2	Sauvignon	36,125,250	0.01%	723	371	202	134	2	94	172	-	89	452	-
i40-3	Cabernet Sauvignon	16,492,150	0.01%	260	145	109	104	38	18	22	-	35	149	-
ERX1002380	Polöskei Muskotaly	35,092,132	0.001%	69	68	35	47	4	50	33	-	14	115	6
ERX1002659	Moscato Rosa	45,170,157	0.001%	102	82	47	57	8	43	25	-	24	159	14
ERX1002378	Lambrusco	40,428,770	0.002%	92	98	56	83	25	59	75	-	22	196	24
ERX1002620	Teroldego	30,029,332	0.03%	2017	1547	517	764	-	-	-	775	403	2429	568
ERX1002377	Alicante Bouschet	29,901,012	0.01%	521	560	290	323	25	196	352	-	144	878	151

* Sample code or SRA number. Samples marked (ds) were analyzed from purified double-stranded RNAs while all other samples were analyzed from total RNA. ^$^ proportion of total trimmed reads mapping to all GaJV-1 references used. ^%^ the genomic RNA reference used is the full-length molecule from isolate i31. ^#^ the genomic RNA reference used is the contig from the isolate Teroldego.

**Table 2 viruses-15-00039-t002:** Non-coding regions (NCRs) and ORF length and encoded proteins molecular weight for all identified genomic RNAs of GaJV-1.

Genomic RNA	5′ NCR (nt)	ORF (nt)	Protein (kDa)	3′ NCR (nt)
RNA1	89	3912	145.8	71
RNA2	97	3060	114.8	73
RNA3	89	1980	72.2	70
RNA4	126	1683	60.7	200
RNA5	97	1377	48.5	387
RNA6	121	1338	49.1	204
RNA7	107	1164	41.7	378
RNArec7-6 *	91 (107) *	1452	52.7	158 (199) *
RNA8	152	1254	44.7	456
RNA9	100	1311	48.2	252
RNA10 *	162 (174) *	1203	43.2	404 (425) *

* For RNArec7-6 and RNA10 from Teroldego, genome ends have not been experimentally extended and verified. The values given for the 5′ and 3′ NCRs are those determined on the assembled contigs, while in parentheses are given full-length NCRs length estimated from comparison with the experimentally determined ends for the various genomic RNAs of the i31 isolate.

**Table 3 viruses-15-00039-t003:** Genome organization and properties of the contigs or scaffolds identified for CaJV-1 and CaJV-2 from public pecan SRAs.

Virus	Genomic RNA	Coverage	5′ NCR (nt)	ORF (nt)	Protein (kDa)	3′ NCR (nt)	% aa Identity #
CaJV-1	RNA1	13x	77	3951	146.7	59	48.3%
	RNA2	12.3x	114	3060	114.9	71	62.7%
	RNA3	13.6x	56	1965	73.1	57	54.8%
	RNA A	14.4x	146	1428	50.3	449	13.9%
	RNA B	21.6x	103	1776	64.5	348	11.3%
	RNA C	15.3x	158	1152	41.6	446	14.4%
	RNA D	14x	na	1402 *	50.7 *	518	13.3%
	RNA9	45x	56	1602	58.6	234	53.2
CaJV-2	RNA1 *	8.8x	na	3532 $	113.7 $	98	45.2%
	RNA2	13.8x	56	3060	113.7	67	62.5%
	RNA3	56.3x	47	1944	71.2	188	54.1%
	RNA9	253.1x	67	1641	60.4	162	51.6%

# Percent aa identity computed with the corresponding protein of GaJV-1. * Contig is 5′ or 3′ truncated so that 5′ or 3′ NCR is missing and ORF 5′ or 3′ truncated. $ scaffold contains an estimated ca. 414 nt internal gap so that encoded protein molecular weight is calculated excluding the gap.

## Data Availability

The sequence of the various GaJV-1, CaJV-1 and CaJV-2 genomic RNAs have been deposited in GenBank and the relevant accession numbers are provided in the text.

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
