# Peer review of "Identification of Seven Additional Genome Segments of Grapevine-Associated Jivivirus 1"

_viruses, 2022, doi:10.3390/v15010039_

Round 1
Reviewer 1 Report
Well-prepared and -written manuscript where a focused while systemic research presented. Switching strategy to from-phloem double-strand RNAs extraction-based high-throughput sequencing from that with total RNAs resulted in the discovery of seven additional genomic segments of the grapevine-associated jivivirus 1 (GaJV-1), which were then investigated in different grape samples for their occurrences, to be long-lasting persistent in vivo other than transitory, despite variable in the presence/absence, and of a recombinant segment representing a typical accidental evolutionary transition. Online searching with the findings of viral and genomic segment counterparts from other plant species thus provided further solid supports to the common presence of such additional segments, assigned later into two novel types of families according to the protein domain homology. Life form of GaJV-1 was confirmed by controlled exclusion of phylloplane fungal contamination to be endophytic in plant, or possible other organisms inside plant, after a previous simple report of graft transmissible of GaJV-1 by Silva et al (2020). Overall, the results are of great values in understanding of biology and evolution of the jiviviruses. Personally, I would be inquisitive to if there is similarity or difference between the seven additional segments and the additional segments found from plants (citrus and grapevine) by Zhang et al. (2022) that was mentioned in the manuscript; of course—if the data is available for comparison.
Author Response
Comments by reviewer #1 have been addressed in the following fashion
Well-prepared and -written manuscript where a focused while systemic research presented. Switching strategy to from-phloem double-strand RNAs extraction-based high-throughput sequencing from that with total RNAs resulted in the discovery of seven additional genomic segments of the grapevine-associated jivivirus 1 (GaJV-1), which were then investigated in different grape samples for their occurrences, to be long-lasting persistent in vivo other than transitory, despite variable in the presence/absence, and of a recombinant segment representing a typical accidental evolutionary transition. Online searching with the findings of viral and genomic segment counterparts from other plant species thus provided further solid supports to the common presence of such additional segments, assigned later into two novel types of families according to the protein domain homology. Life form of GaJV-1 was confirmed by controlled exclusion of phylloplane fungal contamination to be endophytic in plant, or possible other organisms inside plant, after a previous simple report of graft transmissible of GaJV-1 by Silva et al (2020). Overall, the results are of great values in understanding of biology and evolution of the jiviviruses.
Re: we obviously thank the reviewer for his appreciative comments on our work !
Personally, I would be inquisitive to if there is similarity or difference between the seven additional segments and the additional segments found from plants (citrus and grapevine) by Zhang et al. (2022) that was mentioned in the manuscript; of course—if the data is available for comparison.
Re: we agree this would be great. We had in fact this idea when preparing the manuscript and tried to do the comparison. Unfortunately, the sequence information on the contigs analyzed in the Zhang preprint are indicated as provided "in the Data S1" and this "Data S1" is not available on the BioRxiv server. We unfortunately therefore do not see an easy way to perform this comparison.
Please see the attachment

Reviewer 2 Report
The manuscript from Candresse and coauthors present the identification and molecular characterization of new segments of the already reported Grapevine associated jivivirus 1 (GaJV1).
In my opinion the manuscript presents novel data but the feeling while reading it is that authors tried to make conclusions avoiding some strictly necessary analyses.
First of all, while they discuss that most of the Jiviviruses were reported in association to plants, they do not take in consideration that most of these papers, including the one in which GaJV1 was identified, suggest that they probably belong to fungal endophytes. So, since they already have cDNAs from surface sterilized leaves and not treated ones I would appreciate if they can analyze, through qPCR, how this action can influence the detection of GaJV1.
Furthermore, as recently demonstrated the in vitro culture of grape buds (easy and fast to achieve) can lead to the almost complete loss of endophytes. This would further give insights to the real host of this virus.
Finally, I saw that the authors mentioned the paper in which a Jivivirus is reported for Aspergillus. Since raw data from that experiment is available (https://www.ncbi.nlm.nih.gov/bioproject/?term=PRJEB49942) I would appreciate if they can try to use such data to recover the genome segments in the only virus with a recognized host able to support the replication. Probably they will find a high number of reads available and they will elucidate some of the wired results obtained in the low coverage data used.
Author Response
Comments by reviewer #2 have been addressed in the following fashion
The manuscript from Candresse and coauthors present the identification and molecular characterization of new segments of the already reported Grapevine associated jivivirus 1 (GaJV1).
In my opinion the manuscript presents novel data but the feeling while reading it is that authors tried to make conclusions avoiding some strictly necessary analyses.
First of all, while they discuss that most of the Jiviviruses were reported in association to plants, they do not take in consideration that most of these papers, including the one in which GaJV1 was identified, suggest that they probably belong to fungal endophytes. So, since they already have cDNAs from surface sterilized leaves and not treated ones I would appreciate if they can analyze, through qPCR, how this action can influence the detection of GaJV1.
Re: Generally speaking, we feel that this would indeed be a good experiment to perform if trying to solve the issue of whether GaJV-1 is a grapevine or a fungal virus, which is indeed a very worthy question. However, this is not the central question we are addressing in the manuscript, which is the finding that GaJV-1 (and likely Jiviviruses in general) have a very different genomic organization that previously reported, and characterize in a detailed fashion this genomic organization and the GaJV-1 proteins.
Addressing the question of GaJV-1 host would require a novel investigation, with multiple experiments like the one suggested above or the one in the next suggestion, which would necessitate significant time and is therefore in our mind out of the scope of the present manuscript.
Furthermore, as recently demonstrated the in vitro culture of grape buds (easy and fast to achieve) can lead to the almost complete loss of endophytes. This would further give insights to the real host of this virus.
Re: same comment as above. This is a very good idea, but to address a different question than the one we address in the manuscript. We therefore think that engaging in this new question/study is out of scope for the present report.
Finally, I saw that the authors mentioned the paper in which a Jivivirus is reported for Aspergillus. Since raw data from that experiment is available (https://www.ncbi.nlm.nih.gov/bioproject/?term=PRJEB49942) I would appreciate if they can try to use such data to recover the genome segments in the only virus with a recognized host able to support the replication. Probably they will find a high number of reads available and they will elucidate some of the wired results obtained in the low coverage data used.
Re: Another very interesting suggestion, tough we are not sure what the reviewer means by "weird results obtained in the low coverage data used" given that although low, these coverages have allowed us to assemble complete genomic segments and, above all, that it is unclear what the Reviewer considers a "weird" in the results presented. The Aspergillus dataset is not very large, a little under 3.2 million reads but does indeed offer deep assemblies (600-1900x average coverage). By tBLASTx analysis of the novo assembled contigs from this dataset, we identified a total of 5 contigs with homologies to GaJV-1 genomic RNAs. Three match the already known RNA1, 2 and 3 and correspond to the Aspergillus lentulus jivivirus 1 sequences deposited in GenBank. The other two have respectively homologies with RNA4 (e-value 7.4 e-8) and with RNA9 (e-value 5.9 e-43). Interestingly the contig with distant homology to the RNA4, encodes a protein that shares the conserved domain identified in proteins P4, P5, P6, P7, P8 and P10 of GaJV-1 and shown in Figure 4. Similar to GaJV-1 P4 it also possesses a conserved nucleoside ribosyltransferase domain (pfam15891). Taken together these results suggest that the Aspergillus jivivirus possesses also additional genomic RNAs encoding proteins belonging to the two families identified in GaJV-1 (and CaJV-1) and typified respectively by GaJV-1 P9 and by GaJV-1 P4 to P8 plus P10 proteins. On the other hand, the conservation of terminal 5' and 3' sequences between the genomic segments of the Aspergillus virus was less clear than in GaJV-1.However, all five contigs showed an original pattern, with the presence of long incomplete palindromic sequences (between 56 and 295 nt) at their 5' end and of long internal repeats (48 to 88 nt) towards their 3' ends. These interesting results have been added as a few sentences in the discussion and we thank the Reviewer for pointing us toward the Aspergillus dataset, since the results obtained shed indeed as he/she thought a little more light on the biology of Jiviriruses.
Please see the attachment

Round 2
Reviewer 2 Report
I would like to thanks the authors for performing the analyses on Aspergillus.
On the other point, I was referring to the wired results looking at the recombinant segments you found. I can suppose that, give the nature of viruses, it would be quite normal to find recombination or genomic variants. The availability of a deeper sequencing would allow to understand if these recombinant segments are present in a high ratio with the “conventional” genomic segments or if they are present only in small amounts. For this reason I would appreciate if the authors would mention this aspect (the needs of having more sequencing data to understand the nature of such sequences).
Similarly, I asked you about the hosts because you discussed that point in your manuscript. If, as you mentioned in the response, you are not pointing to shed light on the real host I would suggest to remove that part from the discussion.
For all the other aspects the manuscript is clear and well written.
